# The synergy of damage repair and retention promotes rejuvenation and prolongs healthy lifespans in cell lineages

**Barbara Schnitzer**, **Johannes Borgqvist**, **Marija Cvijovic***

Department of Mathematical Sciences, Chalmers University of Technology and University of Gothenburg, Gothenburg, Sweden

* marija.cvijovic@chalmers.se

**Data Availability Statement:** All relevant data are within the manuscript and its Supporting Information files.

## Abstract

Damaged proteins are inherited asymmetrically during cell division in the yeast *Saccharomyces cerevisiae*, such that most damage is retained within the mother cell. The consequence is an ageing mother and a rejuvenated daughter cell with full replicative potential. Daughters of old and damaged mothers are however born with increasing levels of damage resulting in lowered replicative lifespans. Remarkably, these prematurely old daughters can give rise to rejuvenated cells with low damage levels and recovered lifespans, called second-degree rejuvenation. We aimed to investigate how damage repair and retention together can promote rejuvenation and at the same time ensure low damage levels in mother cells, reflected in longer health spans. We developed a dynamic model for damage accumulation over successive divisions in individual cells as part of a dynamically growing cell lineage. With detailed knowledge about single-cell dynamics and relationships between all cells in the lineage, we can infer how individual damage repair and retention strategies affect the propagation of damage in the population. We show that damage retention lowers damage levels in the population by reducing the variability across the lineage, and results in larger population sizes. Repairing damage efficiently in early life, as opposed to investing in repair when damage has already accumulated, counteracts accelerated ageing caused by damage retention. It prolongs the health span of individual cells which are moreover less prone to stress. In combination, damage retention and early investment in repair are beneficial for healthy ageing in yeast cell populations.

## Author summary

Many cell types in nature distribute damage unevenly at cell division resulting in an ageing mother and a rejuvenated daughter cell. Rejuvenation is essential for the viability of a cell population, but it comes at a price: mother cells accumulate damage faster and ageing is therefore accelerated. To elucidate underlying mechanisms of this trade-off, it is crucial to take into account both the single-cell and the population level. Using budding yeast as a model organism, we developed a dynamic model of damage accumulation that allows mapping individual strategies of damage control with properties of the cell population.

**Funding:** This work was supported by Swedish Foundation for Strategic Research (Grant Nr. FFL15-0238 and IB13-0022) to MC. The funders had no role in study design, data collection and analysis, decision to publish, or preparation of the manuscript.

**Competing interests:** The authors have declared that no competing interests exist.

We show how increased damage asymmetry at cell division can ensure the survival of a large fraction of the lineage by diluting the damage over the whole population. At the same time, the cells' healthy lifespans are prolonged if damage repair is particularly effective during early divisions, and thus the burden on the single-cell level is minimised. Consequently, our computational framework allows to concretise terms like rejuvenation and healthy ageing and to combine them to test evolutionary hypotheses.

## Introduction

Even to this day, a conceptual understanding of ageing as a phenomenon is lacking [1, 2]. Accordingly, the task of explaining the origins of specific symptoms of ageing on an organism level beginning from molecular processes in a single cell is highly complicated, and the first step in achieving this goal is to properly define the term. One of the most feasible theories on the origins of ageing, the disposable soma theory [3], states that in circumstances with limited resources an organism will prioritise reproduction over maintenance of the body, called the soma, referred to as a division of labour [4]. Further, its occurrence is motivated by the observation that, in nature, most animals die at a young age due to various factors such as starvation or predation [1], and therefore natural selection will favour genes resulting in rapid growth followed by reproduction as opposed to traits associated with longevity. Thus, the expected consequence of the removal of such factors leading to death is the degradation of the soma since the germ cells are prioritised. To investigate the precise molecular properties of this degradation, it is advantageous to examine the division of labour in less complex biological systems.

The unicellular predecessor of this process can be studied in the budding yeast *Saccharomyces cerevisiae (S.cerevisiae)*. Here, the division of labour occurs on the population level between the larger mother cells, corresponding to the soma, and the smaller daughter cells, corresponding to the germline [5]. In particular, the ageing process in yeast is characterised by an increase in both size and generation time, as well as an accumulation of different types of damage called ageing factors. Some of the most universal ageing factors are misfolded or oxidatively damaged proteins [6–8]. Although the cell has developed intracellular responses to the formation of detrimental byproducts, such as the repair and degradation of damage by an extensive protein quality control system [9–12], these processes are imperfect and under these conditions the accumulation of damage is inevitable [13, 14].

A population level response to the accumulated damage in order to ensure the generation of viable daughter cells is the asymmetric distribution of damage between the progenitor and progeny after cell division. This entails that the mother sacrifices herself by preventing damage from leaking over to the daughter [15]. This asymmetry is a crucial component of ageing, motivated by the fact that it is a conserved mechanism which has been found in species ranging from bacteria to mammals, and in particular it is found in human stem cells [16–24]. The process is essential for maintaining the viability of a population of cells [3, 25–27] by producing rejuvenated daughter cells [6, 28–30]. Simultaneously, over successive divisions damage will accumulate in the progenitor cell which eventually leads to the loss of its fitness and fecundity, i.e. replicative ageing, which can ultimately be measured by the number of produced daughter cells throughout its lifetime. Even though individual mother cells are disadvantaged, it ensures full replicative potential of the progeny and thereby the immortality of the lineage.

Since the budding yeast *S.cerevisiae* exhibits an asymmetric cell division, it has become a successful model organism to investigate replicative ageing [31–34]. Asymmetry in the amount of damage can be achieved by pure size asymmetry between mother and daughter cell, but

yeast cells show various additional damage retention mechanisms that result in an even larger damage polarisation and therefore reinforce rejuvenation [29, 35, 36]. Several studies point to the existence of passive mechanisms that explain the asymmetry based on diffusion arguments and the narrow bud neck from which the daughter grows [37–40]. Even though passive retention can establish damage asymmetry it was also confirmed that there must be additional active mechanisms that lead to aggregate formation and selectively keep damaged or old material within the mother cell compartment at spatially confined spots [41–46], and even transport back damaged proteins from the daughter cell compartment [15, 47]. However, in the late stages of the lifespan of a single mother cell rejuvenation becomes less effective and ageing factors are increasingly passed on to the daughters, which have a decreased replicative potential and, thus, are born prematurely old [48, 49]. Nevertheless, when these prematurely old cells start to proliferate their own daughters, i.e. cells in the second generation, are recovered back to full replicative potential and show an absence of ageing biomarkers, which is called second-degree rejuvenation [48]. To study the impact of these fundamental processes on the ageing process in the context of the disposable soma theory, a population level focus is required.

Together, the repair and retention of damage steer the ageing of individual cells ultimately affecting the well-being of the entire population, but the exact impact of these forces are not understood. To investigate this, single-cell data is not sufficient and information about progenitors is required additionally. Despite advances in experimental techniques, it is not possible to experimentally track all mother and daughter cells in a population over several generations and to gather information about explicit mother-daughter relations. To cope with the vast numbers of cells in a population, replicative ageing due to damage accumulation has therefore been successfully studied with the help of simulations. In that way, the evolutionary benefit of rejuvenation caused by asymmetric cell division has been confirmed [25, 50]. Moreover, using mathematical modelling it was proposed that a more asymmetrical distribution caused by retention of damage is advantageous as a means to cope with high damage formation rates, i.e. stress, and this increases the population fitness [51–53]. Also, an age-dependent maximal degree of retention has been suggested [54]. More recent modelling studies highlight the importance of the repair machinery in replicative ageing since asymmetric damage distribution at cell division alone cannot eliminate damage from the system [55, 56]. However, existing studies have shown the benefit of asymmetric cell division for the population fitness through rejuvenation by measuring average population-based behaviour as the only indicator which entails that the heterogeneity of individual cells in the lineage has not been considered. Typically, the growth rate of the population or its size is considered, but in order to understand the precise underlying mechanisms of how the fitness is increased, properties of individual cells in the lineage have to be accounted for.

Accordingly, the connection between the capacity of the repair machinery or damage retention of individual heterogeneous cells and population level properties, such as the distribution of damage or the overall fitness, remains unclear. To this end, the aim is to investigate the effect of altering the intracellular processes related to ageing of individual cells on features of the population as a whole.

## Results

### From a single-cell to a whole population model of replicative ageing

To investigate the synergistic role of retention of age-related damage and repair of damage on how ageing of the individual cells contributes to the overall behaviour of the entire population and how this affects cellular rejuvenation and health span, we developed a dynamic model

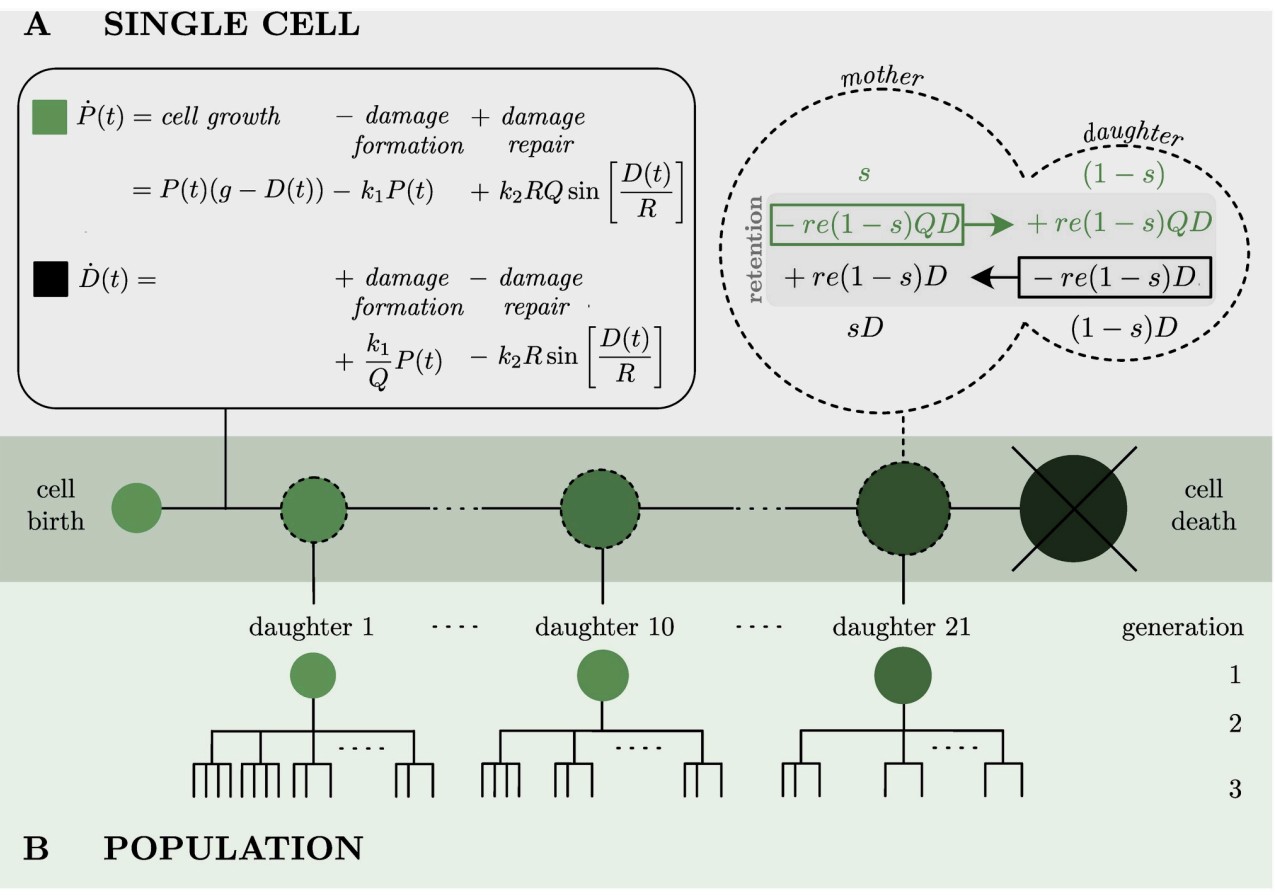

**Fig 1. Schematic view of the model. (A)** Single-cell model in the non-dimensionalised version (see details in S1 Text). Left: Damage accumulation model of single-cells is governed by cell growth, damage formation and damage repair. Right: distribution of functional (green, *P*) and damaged (black, *D*) protein content between mother and daughter cell at division according to the size proportion *s* and the retention factor *re*. The arrows indicate the result of damage retention mechanisms in the model that effectively load the mother cell with more damage. **(B)** Transition to the population model. Each cell division leads to a new-born daughter cell following an independent single-cell model, such that the lineage tree is recursively built up.

based on ordinary differential equations (ODE) that is capable of simulating and mapping replicative ageing on single-cell and population level.

**Single-cell model**: The single-cell model of damage accumulation builds on our experimentally verified replicative ageing model [54], describing the dynamics of functional ($P(t)$) and damaged (malfunctioning) ($D(t)$) proteins over time. In the model presented here (Fig 1A), the protein content evolves non-linearly depending on the growth factor ($g$), the damage formation rate ($k_1$), the damage repair rate ($k_2$), the resilience to damage ($Q$) and a repair capacity ($R$) (see S1 Text for an in-depth description of the parameters and terms in the model). The growth factor and the resilience to damage are essential parts of the single-cell model and have been extensively studied and estimated in [54]. In this work, we focus on damage formation and repair. In particular, we introduced the repair capacity to the model to investigate the effect of accumulated damage on the efficiency of the repair machinery, represented by the repair term $r(D)$ in the dynamic model:

$$r(D) = k_2 R \sin\left[\frac{D(t)}{R}\right], \quad k_2 > 0, \quad D(t) \in [0, 1], \quad R \in [\pi^{-1}, \infty]. \quad (1)$$

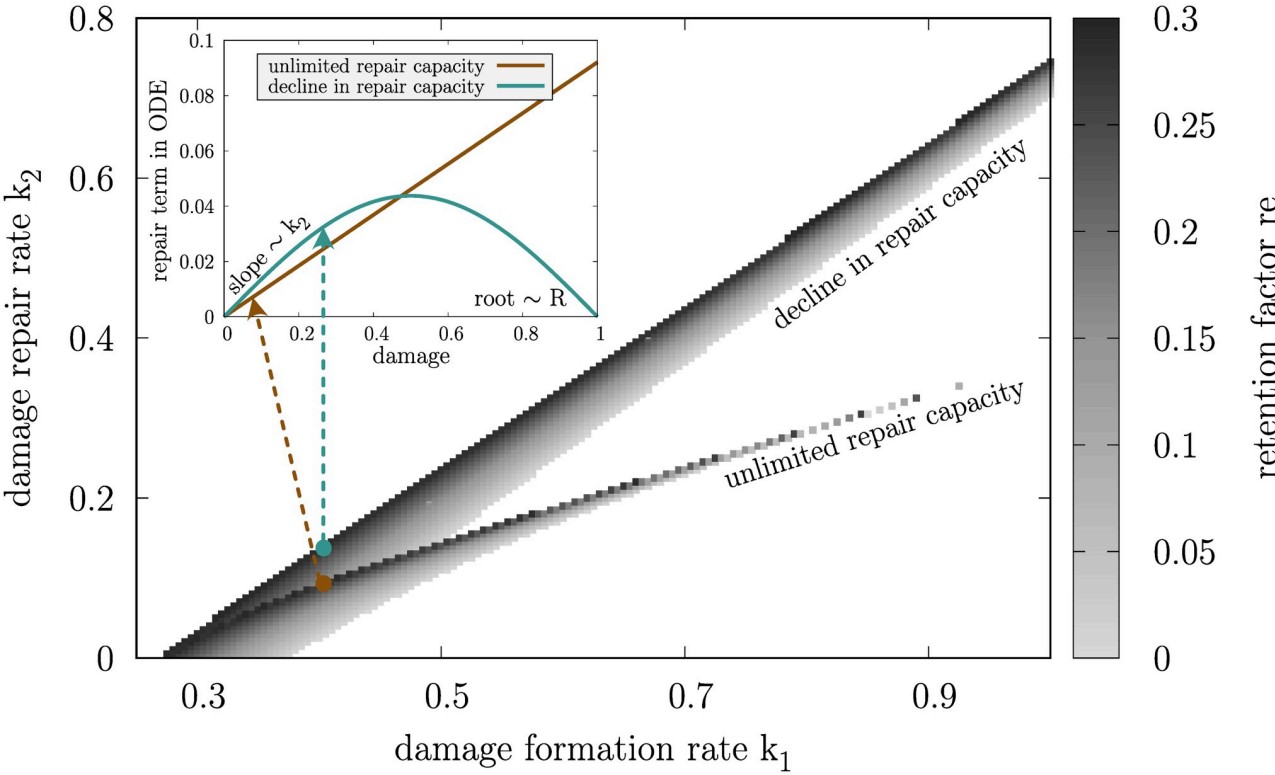

**Fig 2. *Wildtype* yeast parameter set.** Discretised parameter combinations of damage formation rate ($k_1$), damage repair rate ($k_2$) and retention rate ($re$) leading to a constant replicative lifespan of 24 of initially damage-free cells for two distinct cases of repair capacity ($R$): unlimited repair capacity ($R \rightarrow \infty$) and decline in repair capacity ($R = \pi^{-1}$). The parameter space is divided into a grid with $\Delta k_1 = \Delta k_2 = 0.005$ and $re$ is adapted accordingly (S3 Text). The sub-panel shows exemplary repair terms $r(D)$ for constant $k_1 = 0.4$, $re = 0.2957$ and $k_2 = 0.092$ for unlimited repair capacity and $k_2 = 0.138$ for decline in repair capacity.

Repair in the model is therefore steered by the repair rate $k_2$ that determines the rate of repair for young cells with low damage levels, where $r(D) \approx k_2 D$, and the repair capacity $R$ that entails how much the effective repair rate drops towards high damage levels and thus old age (see S2 Text for details). The repair term cannot explain mechanistic details underlying the repair machinery of yeast cells but can help to compare and understand the consequences of specific repair strategies that are still elusive from experimental investigations. In previous ODE models of damage accumulation repair or degradation of damage is typically modelled as linear [51, 52, 54] or saturating functions of the present damage, inspired by Michaelis-Menten kinetics [55, 56]. In the study presented here, linear, saturated and declining profiles can be generated only by varying the repair capacity $R$. Two distinct cases of the repair capacity are considered further in the analysis (Fig 2, repair term in sub-panel): unlimited repair capacity ($R \rightarrow \infty$), corresponding to a constant rate of repair (equivalent to $k_2$) throughout the cell's life independent of the damage levels in the cell since the efficiency of the repair machinery is not influenced by how old and damaged the cell is; and decline in repair capacity ($R = \pi^{-1}$), corresponding to a continuous effect of damaged components on the repair machinery, especially evident towards the end of the cell's life where damage levels are high and consequently the effective rate of repair drops to zero. Intermediate values of $R$ were investigated but do not add to the conclusions of this paper.

During the cell division the protein content is distributed between the mother and the new-born daughter cell according to their size proportion (*s*) and damage retention (*re*), that is based on the principles of active and passive retention mechanisms [15, 37–47] in combination with mass conservation over generations, as in previous replicative ageing models [51, 54, 55, 57] (Fig 1A, see details in S1 Text). Cell division is initiated when a critical amount of functional proteins is reached.

For simplicity, we assume *s* and *re* to be constant over the lifespan of a cell. In the model, we further assume full availability of nutrients at any time such that the total protein amount in the cell grows over time. However, damaged proteins slow down growth and, depending on the repair capacity, also damage repair. In combination with asymmetric damage segregation at cell division, we can simulate damage accumulation in mother cells that leads to the loss of fertility, thus replicative ageing. Subsequently, the cell is considered dead when a critical amount of damage $D_c$ is reached. Both cell division and cell death are modelled as instantaneous events.

The presented replicative ageing model of an individual cell is non-dimensionalised (see S1 Text for details), resulting in the simpler model structure with a reduced number of parameters, thus simplifying the analysis of the system.

**Population model**: From the single-cell model, we recursively build up the lineage by including all new-born cells in the population framework (Fig 1B) with explicit knowledge about all genealogical mother-daughter relations reflected in their lineage positions. After birth, each cell follows its own independent single-cell model that is initialised according to the distribution of proteins at cell division. We account for the individuality of cells by a non-linear mixed effect approach which has been shown to be a powerful tool to include cell-to-cell variability in population analysis [58]. Consequently, in our model the damage formation rate ($k_1$) and the damage repair rate ($k_2$) are drawn from a distribution instead of being constant for all cells. To ensure positivity of the parameters a log-normal distribution was used. Each new cell has therefore a unique set of parameters, composed of deterministic population fixed effects $\bar{k}_i$, $i \in \{1, 2\}$, and random individual mixed effects $\eta_i$:

$$k_1 = \bar{k}_1 \exp[\eta_1], \qquad k_2 = \bar{k}_2 \exp[\eta_2], \qquad \eta_1, \eta_2 \sim \mathcal{N}(0, \sigma^2). \tag{2}$$

All other parameters are specific to a certain population and do not vary across its individuals.

In the course of this work, we introduced a formal measure of rejuvenation (rejuvenation index) as the difference between the cell's and its mother's replicative lifespan scaled by the average replicative lifespan of the population, and a formal measure of health (health span) as the fraction of the cell's lifespan when damage levels are below a given threshold coupled with the number of divisions accomplished until this time point. For the dynamically growing yeast lineage, numerous additional properties of each individual cell and the whole cell population are defined and extracted. Individual cells are characterised by: replicative lifespan, rejuvenation index, health span, generation time, growth per cell cycle, cumulative growth, lifetime, lineage position and generation (Box 1), while the entire population is described by: population size, fraction of rejuvenated cells, fraction of healthy cells and population growth rate (Box 2). Further, distributions of all single-cell properties over the lineage can be studied.

The proposed framework results in a detailed entire cell population model, including all intermediate branches and their explicit relationship, allowing to systematically infer how

## Box 1. Single-cell properties

**P(t)**, **D(t)** represent the content of functional $P$, damaged $D$ proteins in the cell over time. $P(t_0)$ and $D(t_0)$ correspond to the protein content at the cell's birth.

**S(t)** represents the size of the cell $S(t) = P(t) + QD(t)$ over time. The size at cell birth is $S(t_0)$.

**Lifetime** represents the time from cell birth to cell death, $t_d - t_0$.

**Replicative lifespan** $rls$ represents the total number of produced daughter cells before senescence.

**Generation time** represents the time between two successive divisions, $t_n - t_{n-1}$ for $n \in \{1, \ldots, rls\}$.

**Growth per cell cycle** represents the relative size increase between two successive divisions, $\frac{S(t_n)}{S(t_{n-1})}$ for $n \in \{1, \ldots, rls\}$.

**Cumulative growth** represents the relative total size increase during the lifetime, $\frac{S(t_d)}{S(t_0)}$.

**Lineage position** represents the relationship to the cell's ancestors up to the most recent common ancestor of the population (founder cell) represented by a set of indices $\{i, j, \ldots\}$. The cell is the $i^{th}$ daughter of its mother which is the $j^{th}$ daughter of the its own mother and so on.

**Generation** represents the level in the pedigree the cell is in with respect to the most recent common ancestor of the population (founder cell), equivalent to the cardinality of the lineage position.

**Rejuvenation index** $rej$ represents the cell's degree of rejuvenation by $\frac{\Delta rls}{\bar{rls}}$, with $\Delta rls$ being the difference between the cell's replicative lifespan and its mother's replicative lifespan, and $\bar{rls}$ being the average replicative lifespan in the population. A rejuvenated cells has $rej > 0$.

**Health span** $h$ represents the fraction of the cell's lifespan where damage levels are below a critical threshold $D_c = D(t_c)$, defined by $\frac{t_c - t_0}{t_d - t_0} \cdot \frac{rls_c}{rls}$. $t_c$ is defined as the time when the cell reaches $D_c$ for the first time, while $rls_c$ counts the number of divisions until this point. $h \in [0, 1]$. Large values of $h$ correspond to a long health span (S2 Fig).

## Box 2. Population properties

**Population size** represents the amount of cells born in the lineage up to a certain generation.

**Fraction of rejuvenated cells** represents the the percentage of all cells in the population with a rejuvenation index $rej > 0$.

**Fraction of healthy cells** represents the percentage of all cells in the population with a health span $h > h_c$.

**Growth rate** represents how fast the population is growing in the exponential phase.

proposed repair and retention strategies of individual cells alter the propagation of damage in the pedigree and affect cellular rejuvenation and health spans.

## Single-cell replicative ageing is steered by the synergy of damage repair and retention

To investigate how damage formation ($k_1$), damage repair ($k_2$), the repair capacity ($R$) and the retention factor ($re$) contribute to the replicative age of an individual cell, a discrete grid of parameter combinations leading to the average *wildtype* yeast lifespan of 24 divisions [59] was generated for two distinct repair mechanisms: unlimited repair capacity and decline in repair capacity (Fig 2). Here, we assume that cells are damage-free at birth.

Since higher damage formation rates and lower repair capacities are deleterious for the cells, they have to be compensated by either a faster repair or lower damage retention or both together to maintain the replicative lifespan of an individual cell. At the same time, decline in repair capacity corresponds to a decreasing functionality of damage repair when damage levels are high (Fig 2, repair term in sub-panel). In this case, in order to reach the same number of divisions, the cell has to compensate with a more efficient repair rate when damage levels are low, reflected in generally higher values of repair rate $k_2$. Decline in repair capacity leads to increased robustness as more parameter combinations, especially for large damage formation rates, yield the corresponding *wildtype* replicative lifespan. This result suggests that repair is especially important in the beginning of a cell's life when damage levels are low.

In this setting, it is generally not sufficient to tune only one model parameter while keeping other parameters constant, since this immediately alters the replicative lifespan of a cell. Consequently, at least two parameters have to be adapted according to the simulated parameter grid when constraining to *wildtype* cells with the same lifespan. Assigning parameter combinations from the grid to founder cells of cell populations creates similar preconditions for them and at the same time allows to focus on how repair and retention strategies on the single-cell level alter damage accumulation in the resulting cell populations. In addition, the presented parameter grid is in a biologically realistic range and therefore serves as a starting point for the further analysis (see S1 Fig and S3 Text for details).

## Retention leads to bigger populations with lower damage levels

To investigate the effect of damage retention on replicative ageing of the individual cell and the overall population behaviour, *wildtype* yeast populations with different retention factors (Fig 3A) were simulated. The population parameters are chosen according to the previously described grid, where the damage formation rate remained constant, while the repair rate, repair capacity and retention were adapted accordingly. Due to the introduced cell-to-cell variability, respective damage formation and repair rates correspond to the population fixed effect. We found that independent of the repair mechanism, an increase in damage retention leads to larger populations (Fig 3A, population size) with on average lower damage levels at birth (Fig 3A, damage at birth). We further investigated how damage propagates further over successive cell divisions in the population without and with damage retention mechanism (Fig 3B and S7 Fig). The results show that independent of the repair capacity retention lowers the damage at birth in daughters regardless of the replicative age and damage levels of the mother cell. While after successive divisions damage accumulates much faster in mothers that retain damage, during few early divisions both the damage in the mother and the daughter cells stay below corresponding levels of cells where mother and daughter are equally loaded with damage at division, resulting in generally lower damage levels in the population.

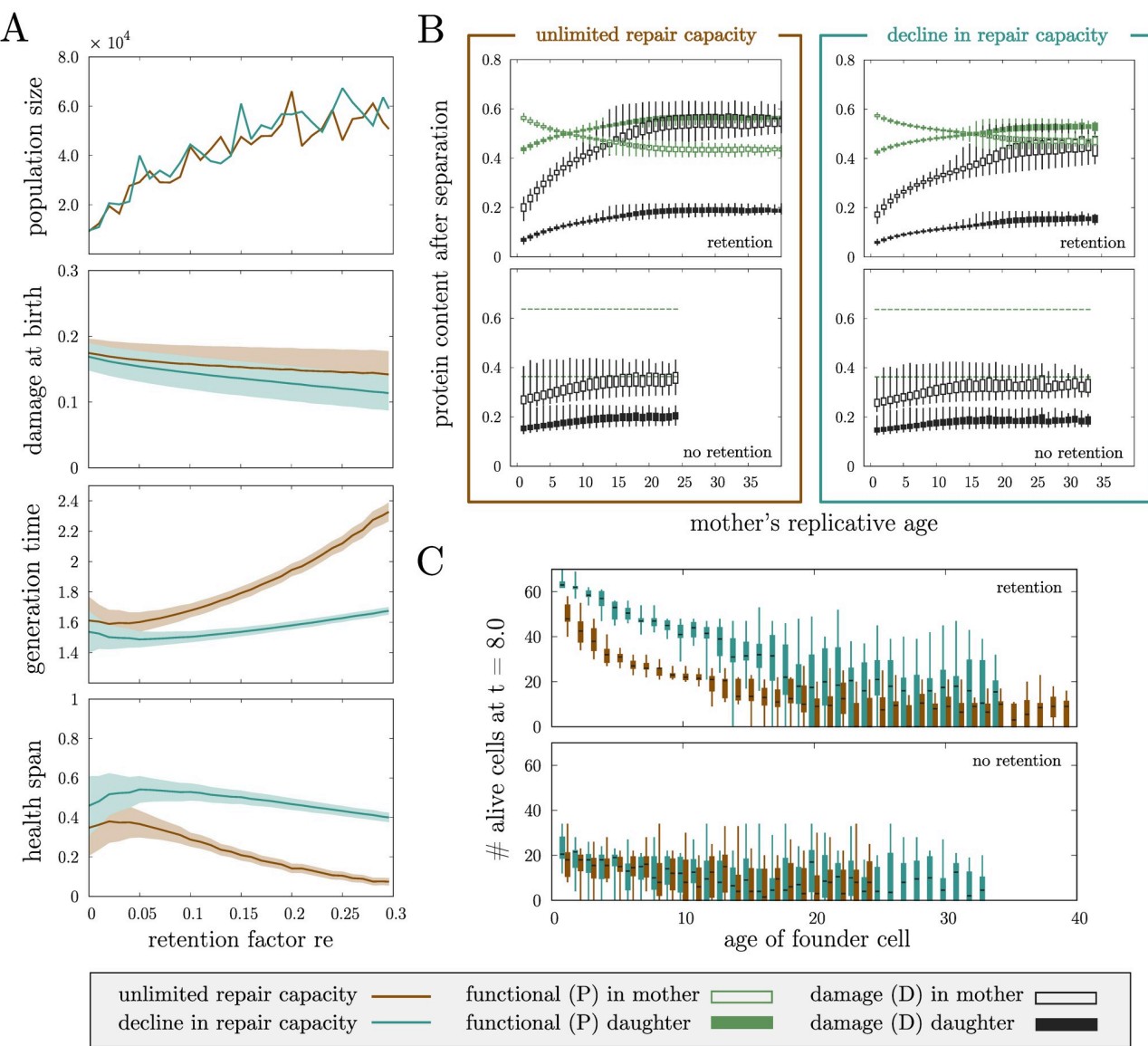

**Fig 3. Single-cell and population properties in *wildtype* yeast populations.** (A) Population size and mean and standard deviation of single-cell properties in *wildtype* cell lineages for varying retention factors *re*. Each population is initialised with 5 independent founder cells with average initial conditions and parameters according to S3 Text. (B) Protein content in mother and daughter cell at division depending on the replicative age of the mother cell. No retention (re = 0.0) corresponds to 64% of the damage in the mother, while retention (re = 0.2957) corresponds to 74% of damage in the mother. The boxplots (whiskers from minimum to maximum) include the whole respective populations that were initialised with 10 independent founder cells with average initial conditions according to S3 Text. A direct comparison between the damage at birth in daughter cells for all four cases can be found in S7 Fig. (C) Number of cells at a specific early time point (*t* = 8.0 in dimensionless time) as an indication of the growth rate of the population in the exponential phase. Each boxplot (whiskers from minimum to maximum) reflects the statistics of 20 populations starting with one founder cell with average initial conditions and parameters according to S3 Text.

We observed that decline in repair capacity generally leads to slightly decreased damage levels at birth compared to unlimited repair capacity (mean damage at birth is reduced by 3% when retention is not present up to 20% with damage retention) (Fig 3A, damage at birth), caused by slower damage accumulation in mother cells (Fig 3B).

To verify that the presented results are not caused by the variability induced by non-linear mixed effects, in particular, the random effect $\sigma$ (Eq 2), we additionally simulated cell populations in which all cells have the same parameter values for damage formation and damage repair rates ($\sigma = 0$), and compared them to populations with respective heterogeneous rate parameters ($\sigma > 0$). This confirms that the effects we observe are not caused by the physical heterogeneity of the cells but by the effect of repair and retention (S8 Fig).

## Retention decreases the variability of the replicative lifespan across the lineage

To understand why populations consisting of cells with active and passive damage retention mechanisms are larger compared to those without, we analysed the cell lineages focusing on four cases: retention and no retention combined with decline in repair capacity and unlimited repair capacity. We used a constant damage formation rate and adapted the damage repair rate in the four cases accordingly (Fig 2). Cells were grouped according to the relation to their grandmother, corresponding to the first two indices of their lineage position $i$ and $j$. The cell is the $i^{th}$ daughter of its mother which is in turn the $j^{th}$ daughter of its own mother (see Box 1 for details).

Without retention mechanisms the population is highly diverse and the replicative age of the mother and even the grandmother at respective cell divisions have great influence on the lifespan of each cell (Fig 4, replicative lifespan). Mostly daughters of young mothers and grandmothers, corresponding to small $i$ and $j$, are able to maintain the replicative lifespan. Daughters of old mothers, i.e. with large $i$, have a substantial disadvantage. In many cases they can only divide few times. Damage polarisation by retention decreases this variability and leads to lifespans around the typical value of *wildtype* cells across the pedigree. All branches in the lineage can quickly recover to full replicative potential such that the whole population can get larger. The repair capacity has in general no influence on the replicative lifespans for a given retention factor.

## Decline in repair capacity results in lower damage levels and prolonged health spans by shortening generation times

The repair capacity profile has an effect on the damage levels in the population, without altering the replicative lifespans. To investigate these mechanisms, properties like generation times and health spans in individual cells were considered (Fig 3A, generation time and health span).

The average generation time is shorter for cells with decline in repair capacity, while the average health span is generally longer. Increasing retention leads to increased generation times and decreased health spans in the populations, for both repair profiles. This is expected since damage retention loads the mother cell with extra damage that inhibits cell growth in the model. However, for decline in repair capacity these alterations caused by retention are only minor such that independent of retention average generation times remain generally short and health spans long. For both repair profiles, the variability is decreased by retention.

This is further confirmed when health span (Fig 4, health span) and generation time (S4 Fig, generation time) were analysed with respect to the lineage position in the population. Both properties are more uniformly distributed across the lineage with retention. Even though there are also cells with very long health span and short generation times without retention, they are restricted to be born from young mothers and grandmothers. The same observation was previously made for the replicative lifespan.

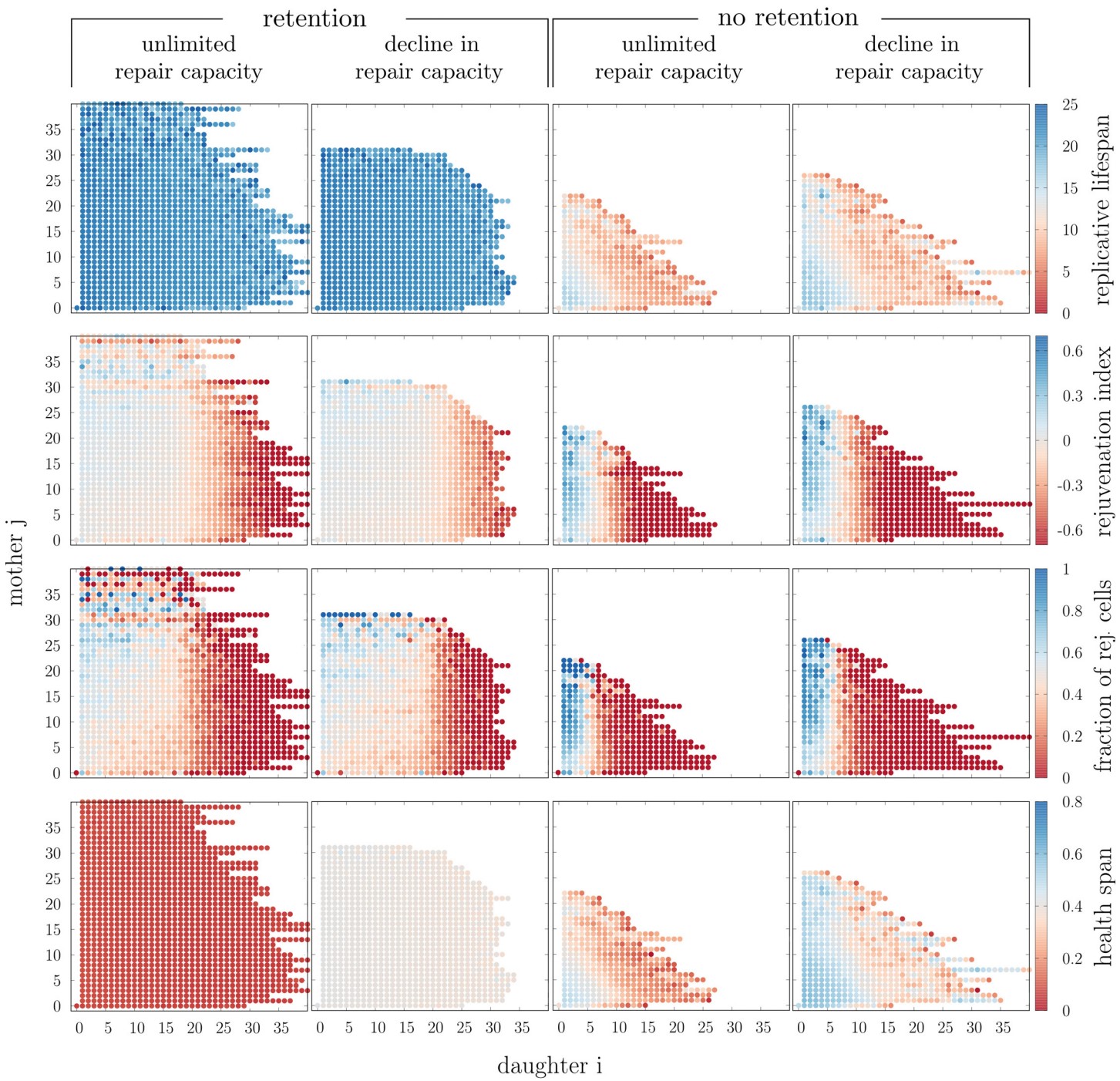

**Fig 4. Importance of the lineage position in *wildtype* yeast populations.** Mean or fraction of single-cell properties for *wildtype* cells that are grouped according to their lineage position with respect to the grandmother (indices *i* and *j* in lineage position). The analysed population is the same as in Fig 3B.

These results can be traced back to the respective repair profiles. Cells with decline in repair capacity efficiently repair damage in early divisions and have therefore short generation times. As a consequence, damage accumulates slower over successive divisions in mother cells after birth (Fig 3B) leading to longer health spans. Damage is diluted faster over the progeny and

daughter cells are smaller at birth (S4 Fig, size at birth). Since the repair is highly affected by damage, it will quickly lose its functionality when damage takes over. Only then damage accumulates fast and proliferation is stopped rapidly. In contrast, unlimited repair capacity leads to faster damage accumulation in mother cells during early divisions and increasing generation times. Yet, cells can cope with damage better towards late divisions. They are able to divide even when already old and thus damaged, resulting in more daughter cells which are born prematurely old.

As already noted, the overall population size is not affected by the repair capacity (Fig 3A, population size), however a fast population growth in the exponential phase is influenced by decline in repair capacity leading to faster growth rates of the population due to decreased generation times, regardless of the age of the population's founder cell (Fig 3C).

## Retention promotes rejuvenation across the cell lineage independent of the repair capacity

We further asked how rejuvenation is affected by retention and repair. We investigated rejuvenation indices across the pedigree for the simulated populations. In particular, we were interested in cells with positive rejuvenation index, i.e. daughters with a longer replicative lifespan compared to their mothers. We observed that rejuvenation occurs independently of the degree of retention and the repair capacity, purely caused by the asymmetry in size at division. Nonetheless, we found that damage retention changes the distribution of rejuvenation across the pedigree for both repair capacities (Fig 4, rejuvenation index and fraction of rejuvenated cells).

Due to the large variability in the lifespans without retention, we found extremely high absolute values of the rejuvenation index in these populations. Rejuvenated cells are however concentrated on specific parts of the lineage: only the first few daughters ($i \leq 5$) have the potential to rejuvenate. If these daughters are from mothers that are born prematurely old, i.e. coming from an old grandmother (large $j$), the daughters nearly certainly rejuvenate and in addition exhibit a large rejuvenation index. Yet, it is a smaller fraction of the whole population that can rejuvenate (S3 Fig, fraction of rejuvenated cells) since later daughters ($i > 5$) have almost no chance to rejuvenate.

As previously described, large variability in replicative lifespans, as observed without retention, are hardly found with retention. As a consequence, rejuvenation becomes much more moderate but instead spreads over a larger part of the lineage. Rejuvenation is no longer concentrated on the few early-born daughters and up to the first 20 daughters of a cell have a significant chance to rejuvenate. Consequently, the total fraction of rejuvenated cells in the whole population increases compared to purely size-dependent segregation of damage.

## Characterisation of rejuvenated cells in the population

To characterise rejuvenated cells further apart from the lineage position, we investigated if they differ in other properties from cells that do not rejuvenate. Our simulated lineages revealed that rejuvenated cells are a small subset of all cells without completely distinct properties. There is a significant overlap between cells with positive and negative rejuvenation index (S5 and S6 Figs).

In particular, we correlated the health span to the number of divisions to investigate if rejuvenated cells have a longer life with lower damage (S5 Fig). In all cases, highly rejuvenated cells exhibit comparably large replicative lifespans. Without retention a high rejuvenation index likely also corresponds to a long health span independent of the repair mechanism. Retention of damage leads to a qualitative difference between the repair capacities. While rejuvenated

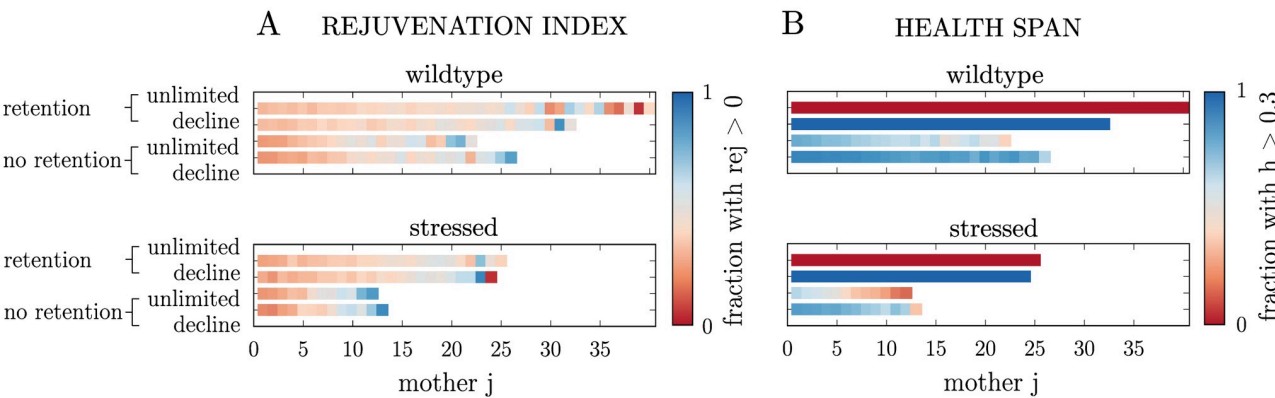

**Fig 5. Stressed *wildtype* yeast populations.** Effect of stress on (**A**) the rejuvenation index and (**B**) the health span in *wildtype* populations. Cells are grouped according to their lineage position *j*. Each population is initialised with 10 independent founder cells with average initial conditions and parameters according to S3 Text. Stress corresponds to an increase in the damage formation rate by 1%.

cells with unlimited repair capacity tend to have short health spans, there is no difference in health spans between cells with decline in repair capacity.

We further correlated the mean generation time of each cell to its size increase per cell cycle to see if rejuvenated cells exhibit different growth behaviour (S5 Fig). In all cases, rejuvenating cells grow comparably little in size per cell cycle. Without retention, cells with large positive rejuvenation index likely have short generation times. With retention the two repair capacities again differ. Unlimited repair capacity correlates long generation times with rejuvenation. In contrast, decline in repair capacity shows again no correlation.

These results suggest that it is in general not possible to predict rejuvenation from single-cell properties disregarding the lineage position. However, there are visible trends for predicting properties given that the cell rejuvenates that are moreover influenced by the repair profiles in case of damage retention.

### Effect of stress on rejuvenation and health

Lastly, we investigated how cell populations are affected by stress (Fig 5) by increasing the damage formation rate $k_1$. Due to accelerated damage formation cells divide fewer times and the populations do not reach the same size in all cases. Rejuvenation can still be observed in the same manner as in *wildtype* cells: mostly cells that come from old grandmothers and comparably young mothers are prone to rejuvenation. The health span is not affected by stress with retention. However, without retention old and damaged grandmothers lead to shorter health spans of their progeny in the second generation.

### Discussion

Starting from a theoretical description of damage accumulation in a single cell, we have developed a holistic model of replicative ageing on the population level. By defining novel features on the population level such as the health span and the rejuvenation index in addition to previously known metrics such as the replicative life span, growth per cell cycle and cumulative growth, we characterised the precise large scale impact of single-cell properties on the lineage. More precisely, we concluded that investing in an efficient repair machinery early in life followed by a steep decline in the capacity to repair later generates longer health spans compared to having a constant repair capacity throughout the lifespan. Further, we showed that retention

of damage, which is detrimental to individual mother cells, reduces the variability in damage levels over generations and increases the average replicative lifespan of the population. Our results emphasise that replicative ageing in a cell population is strongly influenced by how and when damage accumulates during the lives of its individual cells. Thus, using this framework we can concretise vague terms such as healthy ageing and tested evolutionary hypotheses that are otherwise hard to quantify.

Healthy ageing is currently a popular notion in human medicine, partly focusing on an overall good quality of life as long as possible [60]. Based on this concept, we defined the health span $h$ (Box 1) being a quantifiable unicellular metric based on the proportion of an individual cell's life that it remains healthy. In this context, healthy denotes that the intracellular damage levels are below a critical threshold value, i.e. $D < D_c$, such that large values of $h$ imply that the cell spent a large proportion of its life with low damage levels and a small proportion of its life with high damage levels. In addition to this metric, we calculate other properties such as the rejuvenation index, the growth per cell cycle and the cumulative growth, which combined with the individuality of the cells reflected in the rate parameters (Eq 2) constitute a computational setup for analysing ageing on a population level. Using this novel and forceful framework, we can quantify the effect of altering intracellular properties, such as the capacity to repair damage, on the accumulation of damage for entire populations as well as individual cells.

Previously, the importance of an efficient repair machinery has been stressed [55, 56] which lead us to investigate a biologically reasonable repair profile. It is based on the assumption that the capacity to repair damage, in similarity to most other intracellular systems, declines with age. To this end, we introduced a novel repair profile (Fig 2) which invests in repair early in life while no damage is repaired late in life. This was compared to a constant repair capacity throughout the life span of a single cell. A declining capacity to repair damage is not only biologically justifiable but we could show that it is even beneficial for the individual cell as it results in lower generation times and larger health spans compared to the constant counterpart, and in fact this also holds on the population level (Figs 3 and 4). This can be explained by the fact that more damage is cleared early in life, when damage accumulates the fastest, leading to more divisions when the cell has low intracellular levels of damage, while no damage is repaired late in life resulting in the cell spending a small proportion of its life with high damage levels before undergoing senescence. Together with faster division times, the damage is in this case diluted more equally over the population such that the emergent repair force of the whole population removes damage more efficiently from the system.

One of the strengths of the model is that it is not merely restricted to the analysis of individual cells as it has the possibility to extrapolate from the single cell to the population level. With respect to the major metrics of ageing, we showed that retention of damage decreases the variability on the population level. For instance, when studying the rejuvenation index, the replicative lifespan and the health span it is clear that the populations with passive and active damage retention have less variance compared to the counterparts without (Fig 4). This implies that the fate of daughter cells in populations with retention is more similar to that of their mothers and grandmothers compared to populations without retention, where for example some cells have very long replicative lifespans while others have very short ones. The fact that the daughters are more similar to their respective progenitors in populations with retention implies that later branches in the pedigree are more likely to survive which not only increases the overall replicative age but also the size of the population and its growth rate (Fig 3). Furthermore, the variability of the health span is affected in the same manner by retention. With retention, cells in the lineage are similarly healthy independent of the lineage position. In contrast without retention, some cells are much more healthy while other cells have high damage levels already

at birth leading to health spans close to zero, which is extremely dependent on the lineage position.

A relevant question to pose is, under what circumstances is it beneficial and disadvantageous respectively for the progenitor cells to retain damage? In harsh environments characterised by highly variable conditions cells die from other factors than ageing [1], suggesting it could be advantageous to have a few cells with high fitness which would argue in favour of cells not retaining damage. However, in more stable environments where food is more plentiful retention of damage could be beneficial as populations become bigger and the average fitness is higher. Moreover, this would suggest that ageing has evolved as a consequence of organisms moving to nutrient-rich conditions with little stress.

## Materials and methods

All simulations and their analysis were performed in the programming language Julia [61] and were run on a computer with an Intel(R) Xeon(R) Platinum 8180 CPU @ 2.50GHz and Julia version 1.1. The developed model can be downloaded from github https://github.com/cvijoviclab/RejuvenationProject. Model parameters, constraints and relevant pseudo code for creating lineages can be found in S3 Text.

## Supporting information

**S1 Fig. Sensitivity of population properties to parameter choice.** Single-cell (mean and standard deviation) and population properties of populations as a function of the damage formation rate $k_1$ for different repair capacities $R$ and retention factors $re$. The damage repair rate $k_2$ is chosen according to the surface for *wildtype* cells (Fig 2) and each population is initialised with 10 independent founder cells with average initial conditions and parameters according to S3 Text. In general, the qualitative results from this paper remain the same independent of $k_1$. Unlimited repair capacity is more affected by altered damage formation rates than decline in repair capacity underlining once more that repair is especially important in the beginning of the life and moreover leads to more robustness. The dashed line indicates $k1 = 0.4$ underlying many simulations in the paper, since it is one of the smallest where retention and no retention can be compared, and further the cells have comparably short lifetimes reducing the computational cost of the simulations.
(EPS)

**S2 Fig. Effect of the choice of the threshold $D_c$.** Mean health span as a function of the lineage position $i$. Each population is initialised with 10 independent founder cells with average initial conditions and parameters according to S3 Text. The threshold for the health span is chosen as $D_c = 0.5$. Too high and too low values are not reasonable to take, since then all cells have either health span of $h \approx 1$ or $h \approx 0$ respectively. The qualitative conclusions in this paper hold independent of the choice of $D_c$ if not too close to the extremes.
(EPS)

**S3 Fig. Complementing population averages.** Mean rejuvenation index, replicative lifespan and growth per cell cycle with standard deviation and fraction of rejuvenated cells in *wildtype* cell lineages for varying retention factors re. The analysed populations are the same as in Fig 3A.
(EPS)

**S4 Fig. Complementing single-cell properties depending on the lineage position.** Mean generation time, growth per cell cycle, damage and size at birth for *wildtype* cells that have the

same lineage position with respect to the grandmother (indices $i$ and $j$ in lineage position). The analysed population is the same as in Figs 3B and 4.
(EPS)

**S5 Fig. Correlation between single-cell properties.** Correlation between health span and replicative lifespan and between mean generation time and mean growth per cell cycle in *wildtype* populations with focus on rejuvenated cells. Each point corresponds to a cell in the lineage. Points can lie on top of each other. The analysed population is the same as in Figs 3B and 4.
(EPS)

**S6 Fig. Correlation between single-cell properties.** Correlation between the lineage position $i$ and replicative lifespan and between aize at birth and cumulative growth in *wildtype* populations with focus on rejuvenated cells. Each point corresponds to a cell in the lineage. Points can lie on top of each other. The analysed population is the same as in Figs 3B and 4.
(EPS)

**S7 Fig. Damage distribution at birth in daughter cells depending on mother age.** Boxplot (whiskers from minimum to maximum) of the damage distribution in daughter cells depending on the age of the respective mother cell for the extreme retention factors and repair capacities. The analysed population is the same as in Figs 3B and 4.
(EPS)

**S8 Fig. Effect of non-linear mixed effects on populations.** Mean and standard deviation or absolute values of several single-cell and populations properties for varying retention factors $re$ in *wildtype* cell lineages with ($\sigma = 0.005$) and without ($\sigma = 0.0$) varying rate parameters according to Eq 2. Naturally, the populations with equal parameters have a decreased variability in properties like the replicative lifespan and the rejuvenation index. Other features, as the population size, generation times, growth, health span and initial damage, do not vary significantly. Only the fraction of rejuvenated cells shows qualitative differences. While the fraction of rejuvenated cells in the homogeneous population decreases with increasing retention of damaged proteins, it can be maintained at a higher level ($\approx 40\%$) if cells in the population show variability in $k_1$ and $k_2$. It is a direct consequence of an increased variability in replicative lifespans caused by the mixed effects that are especially evident for high damage retention. In contrast, in this regime almost all cells in the population with $\sigma = 0$ have the same replicative lifespan such that rejuvenation (positive rejuvenation index) is less likely. These results confirm that mixed effects have no further influence on the population fitness in our model, and results on rejuvenation and health in the cell lineages are not caused by the physical heterogeneity of the cells but by the effect of repair and retention.
(EPS)

**S1 Text. Model construction and non-dimensionalisation.** Detailed information about the single cell model as well as the non-dimensionalisation of it.
(PDF)

**S2 Text. Repair term with the repair capacity R.** Detailed description and justification of the repair term with special focus on the repair capacity $R$.
(PDF)

**S3 Text. Computational guidelines.** Parameter choices and pseudo-code for relevant computational procedures.
(PDF)

## Acknowledgments

We would like to thank all past and present members of the CvijovicLab for valuable input and careful reading of the manuscript.

## Author Contributions

**Conceptualization:** Barbara Schnitzer, Johannes Borgqvist, Marija Cvijovic.

**Formal analysis:** Barbara Schnitzer.

**Supervision:** Marija Cvijovic.

**Writing – original draft:** Barbara Schnitzer, Johannes Borgqvist.

**Writing – review & editing:** Marija Cvijovic.

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
