## [Decision Letter · Decision Letter 0]

25 May 2020

Dear Dr. Cvijovic,

Thank you very much for submitting your manuscript "The Synergy of Damage Repair and Retention Promotes Rejuvenation and Prolongs Healthy Lifespans in Cell Lineages" for consideration at PLOS Computational Biology.

As with all papers reviewed by the journal, your manuscript was reviewed by members of the editorial board and by several independent reviewers. In light of the reviews (below this email), we would like to invite the resubmission of a significantly-revised version that takes into account the reviewers' comments. In particular, the authors must demonstrate how changes the underlying assumptions and functional terms in their model affect their conclusions and clearly justify the parameters and assumptions used in their models. Once all this is done, it remains to be seen if the revised manuscript is able to meet the threshold of biological significance for this journal. 

We cannot make any decision about publication until we have seen the revised manuscript and your response to the reviewers' comments. Your revised manuscript is also likely to be sent to reviewers for further evaluation.

Sincerely,

Oleg A Igoshin

Associate Editor

PLOS Computational Biology

Jason Haugh

Deputy Editor

PLOS Computational Biology

Reviewer's Responses to Questions

**Comments to the Authors:**

Reviewer #1: The paper developed a dynamical model for damage accumulation in budding yeast cells including the production of damaged proteins and the asymmetric segregation of proteins at division quantified by the retention factor. If I understand correctly, the paper's main conclusion is that damage retention and early investment in repair are beneficial for healthy ageing in yeast cell populations. My main concern is that the dynamical model appears to be very specific, especially in terms of the mathematical form of the repair capacity. It appears that many of the conclusions of this paper relies on the particular forms they choose and the generality of their results are not justified. Moreover, the paper is not clearly written and some of the formulas and results are not explained. Below are some more minor comments.

(1) It is unclear what is "r" in Eq.1 and it is also unclear why the authors choose sin function which can lead to negative value.

(2) The right panel of Fig.1A is unclear, for example, the meaning of black and green arrows are not explained.

(3) The meaning of the resilience to damage Q is not explained but it looks quite important as it appears both in the dynamics of protein level and the retention of protein at cell division.

(4) In the section of population model, it's not clear what is the "explicit knowledge about all mother-daughter relations". It looks like k1 and k2 are drawn independently between mother and daughter celle.

(5) Line 149-154 on two distinct cases of the repair capacity are unclear.

(6) Line 196-198 on the discussion of Figure 3B argues that the damage at birth in daughters is lower when there is retention. The authors may want to show the results with retention and without retention in the same figure to be more clear.

(7) On line 184, it is unclear how the population size is defined as it changes with time. I guess the authors are computing the population size after a constant time starting from the same initial condition.

(8) It is unclear how unlimited repair capacity leads to increased generation time.

Reviewer #2: Schnitzer et al. developed a dynamic model of replicative ageing in studying damage accumulation in a single cell. Through the modeling approach the authors showed that active damage retention lowers damage levels in the population by reducing the variability across the lineage. They also found that repairing damage efficiently in early life as opposed to intervention in the later stage could slow ageing progress caused by damage retention. The metric “health span” investigated in this study is also interesting which could be used to quantify topics related to healthy ageing and quality of life.

My main concern is about the generality of the modeling results. More specifically, the conclusions of this study were mainly made based on one single set of parameter values shown in Table S2. The authors should provide justifications about how those parameter values are chosen. For parameter values without experimental/literature backup, the authors should also investigate and report how the results and conclusions would be impacted/altered by conducting parameter sensitivity analysis.

In Fig. 1A, authors should address the rational behind the first term P(t)(g – D(t)), and why there is no rate constant associated with this term.

Minor point, in Line 301, “generation time of each cell to it size increase”, it looks like there is a typo here. The authors may want to write “its” instead of “it”.

**Have all data underlying the figures and results presented in the manuscript been provided?**

Reviewer #1: None

Reviewer #2: Yes

PLOS authors have the option to publish the peer review history of their article (what does this mean?). If published, this will include your full peer review and any attached files.

Reviewer #1: No

Reviewer #2: No
---

## [Decision Letter · Decision Letter 1]

3 Jul 2020

Dear Dr. Cvijovic,

Thank you very much for submitting your manuscript "The Synergy of Damage Repair and Retention Promotes Rejuvenation and Prolongs Healthy Lifespans in Cell Lineages" for consideration at PLOS Computational Biology.

As with all papers reviewed by the journal, your manuscript was reviewed by members of the editorial board and by several independent reviewers. The manuscript was sent to two original reviewers who are mostly satisfied with the revisions but based on the comments to the first submission, we have also decided to solicit an opinion for another reviewer with specific expertise. As you can see below, there are several important revisions that the additional reviewer is requesting. In light of the reviews (below this email), we would like to invite the resubmission of a significantly-revised version that takes into account the reviewers' comments.

We cannot make any decision about publication until we have seen the revised manuscript and your response to the reviewers' comments. Your revised manuscript is also likely to be sent to reviewers for further evaluation.

Sincerely,

Oleg A Igoshin

Associate Editor

PLOS Computational Biology

Jason Haugh

Deputy Editor

PLOS Computational Biology

Reviewer's Responses to Questions

**Comments to the Authors:**

Reviewer #3: Summary:

Schnitzer et al. introduce a dynamical model which helps study the effect of intracellular processes at a population level. They also define interesting new measures as the rejuvenation index and the health span that help analyze some of the characteristics of budding yeast cells at a population level. They find evidence of a trade-off between damage repair and damage retention to replicate replicative life-spans in WT cells. They find that increased retention leads to larger, healthier populations and decreases the variability of the replicative life-span of offspring. Moreover they find that a decline in repair capacity reduces damage and improves health in contrast to unlimited (constant) repair capacity.

General Comments:

The main interest of the paper; using mathematical and computational analysis to understand how intracellular processes affect observables at a population level is innovative and interesting and has not yet been put to use thoroughly in existing bibliography (to my knowledge). The potential to this approach is vast and can have very strong implications as it provides new ways of analyzing feasible experimental data and its relation to intracellular processes which are difficult to study or analyze directly. The first main result, that active damage retention lowers damage levels, is not very striking and has been discussed in a fair amount of papers in the past ten to twenty years. The second main result, that repairing damage in early life can be more effective than continuously and constantly throughout life, is more captivating as it links to existing theories of ageing (the disposable soma theory) and to existing experimental results.

The introduction of new measures at a population level is very promising and can help reduce the gap between existing theories of intracellular processes and experimental observables.

Although the methods and some of the results can be of interest in the community, I agree with the previous referees that the model seems too specific and that the conclusions rely strongly on the functional forms chosen (more details on this below). Furthermore, comparisons or referrals to available published experimental work are very limited. Improving this aspect could have important consequences in terms of validating the simulations’ results and reducing the degrees of freedom in the models’ dimensional space. Moreover, I believe that the use of the term “active” when referring to retention can be misleading (see detailed comment below) and that the equations used to describe intracellular processes remain unjustified and are not necessarily in line with the existing mathematical theory, which decreases the impact of the paper’s discoveries.

Major points:

1. The equations used for the intracellular dynamics in the model are still not fully justified. In particular the equations concerning the distribution of damage after cell division. These equations don’t seem consistent with more explicit existing models of aggregate dynamics and with the theory of escape times of diffusing particles inside confined domains (see work from Holcman and Schuss). Also, are these equations based on a 2 or a 3 dimensional representation of the cell? It should be at least explained why this is different or how it relates to them (if it improves them, focuses on something else etc.). See for instance:

A. Kinkhabwala, A. Khmelinskii, and M. Knop, BMC biophysics 7, 10 (2014),

M. Andrade-Restrepo, Biophysical journal 113, 2464 (2017),

C. Paoletti, S. Quintin, A. Matifas, and G. Charvin, Biophysical journal 110, 1605 (2016). Moreover, and as explained below, significant retention can occur as well due to passive effects, hence claiming that the retention term in the equation is only a consequence of active processes is not necessarily true.

The explanation for the sine in the damage repair term given to the previous reviewer should be added to the paper. Otherwise, it seems artificial. More minor comments on these equations are below.

2. As anticipated above, the use of the term “active” is misleading in the paper. While the degree of retention is certainly varied throughout the paper and it can have a strong impact on the results, it is never specified to what degree this retention occurs due to active processes (see previous point as well). Passive processes (confined diffusion, crowded environment, aggregate fusion etc.) can also justify much of the retention observed in real life cells even if the presence of Active-quality-control seems more and more likely. Therefore, while retention is undoubtedly important, the effect of active processes can or can not be significant in the observed results and has not yet been properly analyzed to be able to claim that active retention and not just retention is behind the paper’s discoveries.

Again, since there is no reference to more explicit and detailed models of intracellular dynamics it seems difficult to justify why retention in the model must be active.

3. Comparisons to available experimental data are very limited. I believe some of the results could be compared with experimental papers concerning replicative life-spans of budding yeast cells and damage accumulation (proportion of damage in mother and daughter cells) and of 2nd-degree rejuvenation. As one reviewer pointed out before, experimental validation or comparisons with experimental data of parameter values could improve the feeling that the equations used are too “artificial” and that results strongly depend on their form. For instance, a decline in repair capacity yields a higher k_2 (Fig. 2) which could alone justify the lower damage levels observed. So the effect would not come from a better strategy but form a higher maximal repair rate.

Minor points:

1.There are still some spelling mistakes and misuse of sentence connectors which affect readability.

For instance, in the author summary it says: “a model organisms”. The subsequent sentence is also strange, particularly the “between how”. Could be something like: “...map individual strategies of damage control with properties of the cell population”. Also on line 274 from page 9 it says: “populations” rather than “population”

2. I really don’t see the point in introducing the parameter Q in the model. I understand that this parameter was analyzed in previous works by the authors but in the context of this paper it does not seem to serve any purpose and could be incorporated as part of the other terms. If the term is however essential for the authors’, it should be justified better why it is included and how it affects the observed results. When it comes to mathematical modelling, simpler models are more powerful and less prone to bias.

3. Bibliography is a little outdated. Specially concerning active quality-control mechanisms in budding yeast. See for instance J. Saarikangas, F. Caudron, R. Prasad, D. F. Moreno, A. Bolognesi, M. Aldea, and Y. Barral, Current Biology 27, 773 (2017), J. Saarikangas and Y. Barral, Elife 4, e06197 (2015). I believe a general review of recent bibliography on the subject together with some minor revisions and some rephrasing (while extending relations to other work) could only improve the paper’s impact in the field.

4. Also, I don’t see how generating k_1 and k_2 randomly rather than taking them constant impacts positively the results rather than just adding unnecessary noise. Does this change the results qualitatively? If not it could just be put as a comment in the paper with a proper analysis in the appendix. Again, if it is essential it should be justified.

5. There is existing experimental data on the rates of formation of damaged proteins; see for instance J. Saarikangas and Y. Barral, Elife 4, e06197 (2015). This could help reduce the model’s degrees of freedom.

6. Active quality control does not only refer to active transport (page 2). There are other possible mechanisms of damage retention that don’t involve active transport of aggregates. See for instance R. Spokoini, O. Moldavski, Y. Nahmias, J. L. England, M. Schuldiner, and D. Kaganovich, Cell reports 2, 738 (2012).

7. In Fig. 3B. The top two green lines are not clearly explained. What does “intact in mother/daughter mean?”

8. I am not sure that the assumption of having constant damage retention is very consistent with real-life processes. The choice of this assumption is also not properly justified.

9. Fig. 3A top seems to reach a plateau, meaning that too much retention does not increase population size.

10. The term “neglected” in page 2 is a bit strong, perhaps it could be changed to “not considered”.

**Have all data underlying the figures and results presented in the manuscript been provided?**

Reviewer #1: None

Reviewer #2: Yes

Reviewer #3: Yes

PLOS authors have the option to publish the peer review history of their article (what does this mean?). If published, this will include your full peer review and any attached files.

Reviewer #1: No

Reviewer #2: No

Reviewer #3: No
---

## [Decision Letter · Decision Letter 2]

4 Sep 2020

Dear Dr. Cvijovic,

We are pleased to inform you that your manuscript 'The Synergy of Damage Repair and Retention Promotes Rejuvenation and Prolongs Healthy Lifespans in Cell Lineages' has been provisionally accepted for publication in PLOS Computational Biology.

Best regards,

Oleg A Igoshin

Associate Editor

PLOS Computational Biology

Jason Haugh

Deputy Editor

PLOS Computational Biology

---

## [Editor Report · Acceptance letter]

6 Oct 2020

PCOMPBIOL-D-20-00614R2 

The synergy of damage repair and retention promotes rejuvenation and prolongs healthy lifespans in cell lineages

Dear Dr Cvijovic,

I am pleased to inform you that your manuscript has been formally accepted for publication in PLOS Computational Biology. Your manuscript is now with our production department and you will be notified of the publication date in due course.

With kind regards,

Sarah Hammond
